# Time-Lapse Camera Monitoring and Study of Recurrent Breaching Flow Slides in Cap Ferret, France

Yves Nédélec * , Philippe Fouine, Cyrille Gayer and Florent Collin

Cerema Sud-Ouest, 24 Rue Carton, CS 41635, CEDEX, 33073 Bordeaux, France; philippe.fouine@cerema.fr (P.F.); cyrille.gayer@outlook.fr (C.G.); florent.collin@cerema.fr (F.C.)
* Correspondence: yves.nedelec@cerema.fr

**Abstract:** In this paper, we present a low-cost method designed to monitor recurrent breaching flow slides that impact the security of a beach. This beach, located in France at the inlet of Arcachon Bay, connects a sand spit to a tidal channel while ending at the toe of a coastal defense. Monitoring is based on capturing images and intends to add continuous information to intermittent direct observations so that triggering and influencing factors can be assessed more precisely. The method is based on time-lapse picture collection and processing. The field of view shows successive emerged manifestations of flow slide phenomena, as well as some possibly related environmental elements. On-site application for 576 days provides important indications and details on flow slide event progress and beach recovery. A simple but quantitative analysis of the influence of sand spit topographic changes is proposed as a preliminary approach of the method's suitability for studies of environmental processes in conjunction with coast protection.

**Keywords:** coastal erosion; breaching flow slide; tidal inlet; sedimentary process; coastal defense; beach stability; low-cost

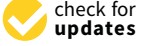



## 1. Introduction

Cap Ferret, located on the southwestern coast of France in the Gironde department, is a natural sand spit stretching between the outlet of an estuarine lagoon called Arcachon Bay (French: Bassin d'Arcachon) and the Atlantic Ocean (Figure 1). Its southern end splits the shoreline into an armored section, heading northward along the tidal channel and the Atlantic shoreline. Since the erection of defenses along the tidal channel, brutal slumps have appeared due to significant processes of coast and bank erosion [1]. Since the southern groyne was constructed in 1995, the sandy beach next to these defenses has also been subject to erosion; many of these slumps have been caused by both wave action and tidal currents that may reach high velocities at ebb time. Consequently, these have shown continuous changes in the shape and state of compaction. These slumps have been identified as breaching flow slides, also called retrogressive breach failures. For a few decades, they have often occurring twice per week, or more. The frequency of these events and the dangers they represent to walkers or fishers may lead to local authorities prohibiting access to the beach and seeking to improve their knowledge about this situation.

Coastal flow slides or retrogressive breach failures are uncommon forms of erosion events that impact compact underwater sandy or silty slopes. They mainly occur in tidal environments, but can be observed on riverbanks or lake shores as well. Their course is driven by gravity, but counter-intuitively, they progress upward. The complex process is triggered by processes at the toe, due to various identified factors (groundwater outflow, dredging, and bed incisions). The release of sediment particles from a vertical wall by sand dilatancy is activated. The wall retrogrades onshore at a slow and steady rate of approximately 0.8 m per minute, and can take the form of a circular excavation in the beach [2–4]. It grows in height and circumference as a plume of sand is drawn away by

density currents (Figure 2). The wall may fail with spectacular collapses and scars [5], which are potentially a deadly threat to beach walkers or fishers, such as in the case of North Wildwood, New Jersey [6]. The phenomenon may seriously endanger nearby constructions too, and further studies are needed in order to properly perform hazard assessments [7].

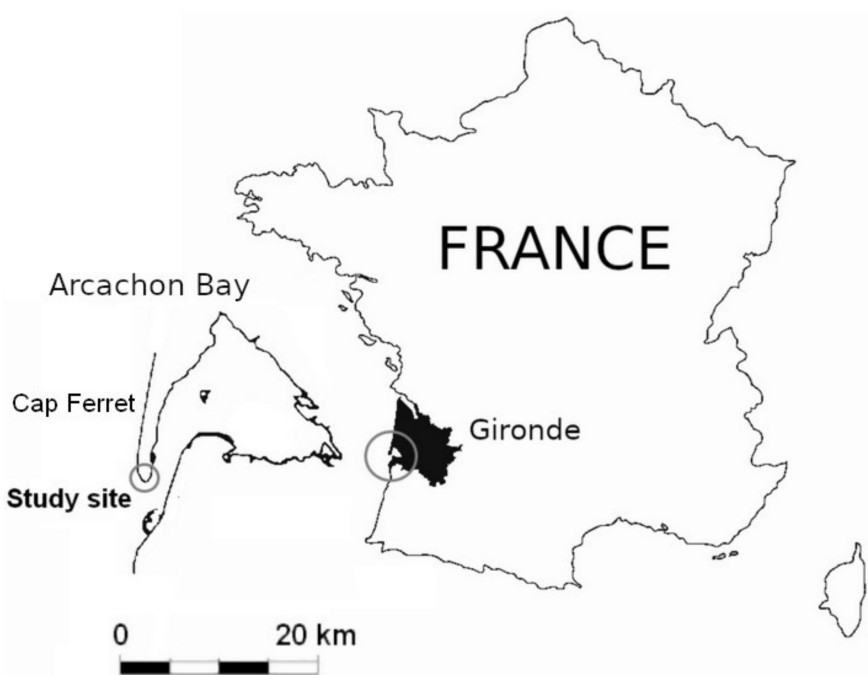

**Figure 1.** Map of France and the Gironde department, indicating the location of Cap Ferret and the study site along the coastline of Arcachon Bay.

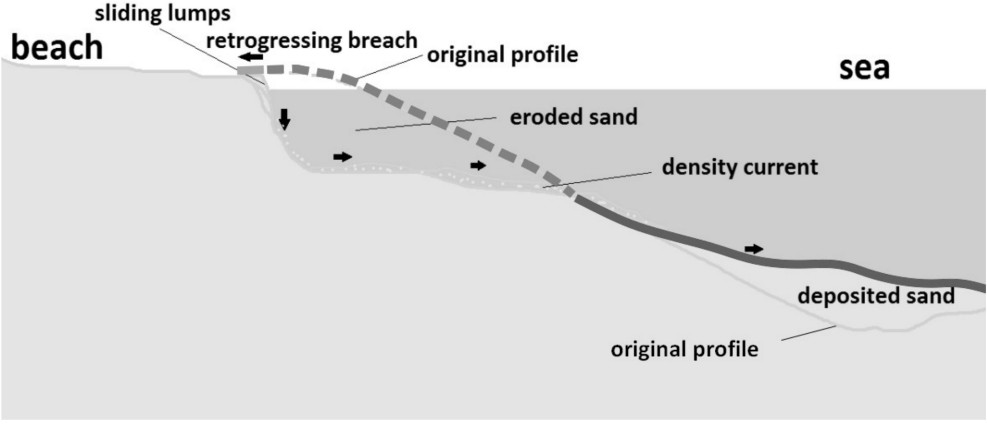

**Figure 2.** Schematic side view of flow slide process.

Improvements in current knowledge and modeling about coastal flow slide dynamics [8,9] have used several approaches. Laboratory experiments explore the physical mechanisms and factors involved [10]; extensive site-scale surveys and investigations enable the linking of breaching events to the morphological evolutions of estuaries [11,12]; and finally, field observations at predisposed locations bring information on the course, size and consequences of locally recurrent events [4,13–16].

However, the randomness and mainly underwater character of this process make assessments of it as an environmental hazard difficult, even in predisposed locations.

Therefore, a new method for coastal flow slide monitoring was conceived at the southern end of Cap Ferret in order to inspect its superficial and emerging dynamics, together with its interactions with surrounding environmental features.

The main objective of this method was to continuously observe beach shape and flow slide events so that frequency could be assessed. A second objective was to capture details on event progression over a short time interval. The experiment was designed with low-cost and robust equipment because of possible damage caused by impacts of waves or by shocks with floating objects.

The study site, the experimental setup and the monitoring procedure are described in this paper, followed by preliminary and qualitative results on flow slide dynamics studied as an event-oriented chronology.

## 2. Study Site

The experimental site was located on the southern beach of Cap Ferret, on the south-west Gironde coast in France (Figure 1). A camera was set up to capture views of flow slides always located near the toe of the southern end of a riprap groyne field bordering the tidal channel on the east. The monitored area of this beach extended from this groyne to a small sand spit facing the Atlantic Ocean further west, enabling environmental observations. Figure 3 shows a map of this monitored area, together with some topographic details.

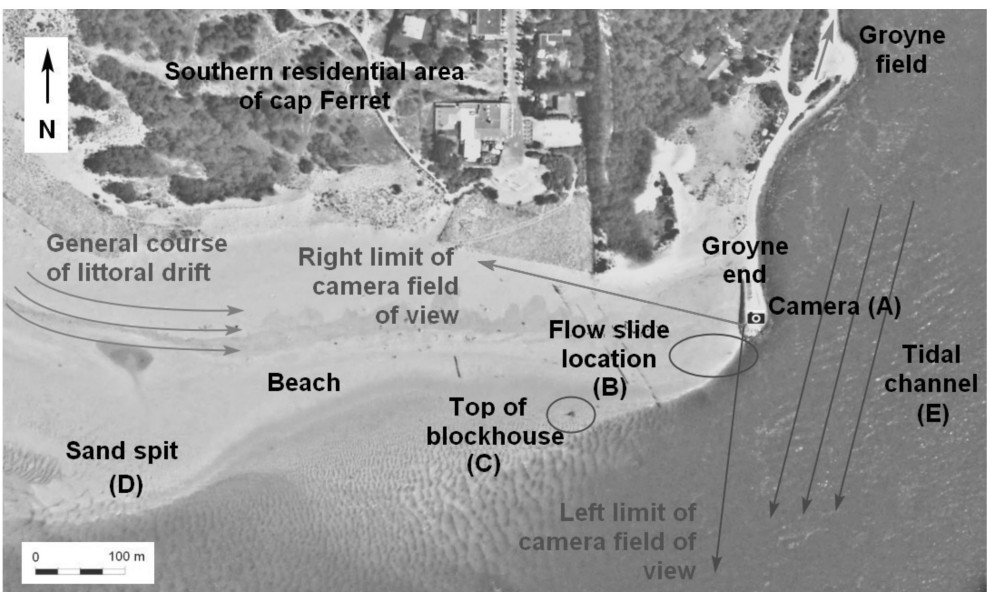

**Figure 3.** Map of the monitored beach centered towards the southwest, and environmental points of interest: (**A**) riprap in which the camera was inserted; (**B**) area exposed to flow slides; (**C**) top of a blockhouse from World War II; (**D**) western sand spit; and (**E**) outlet of tidal channel (background aerial view: Géolittoral Ortho-Littoral V2 © Cerema-IGN).

Five sedimentary, hydrodynamic and environmental points of interest can be found at the study site:

1.  Blocks forming the eastern groyne where site observation was performed and near to the area susceptible to flow slides;
2.  The area susceptible to flow slides itself;
3.  An immovable World War II blockhouse partially stuck in the sand since, showing a corner pointing upside, easy to spot, and providing a good sand level indicator;
4.  The western end of the beach exposed to ocean waves, shaped like a moving sandy hook;
5.  The outlet of the tidal channel, often marked by turbulence and strong flow patterns.

Figures 4 and 5 show the site from the water surface and from the top of the groyne, respectively.

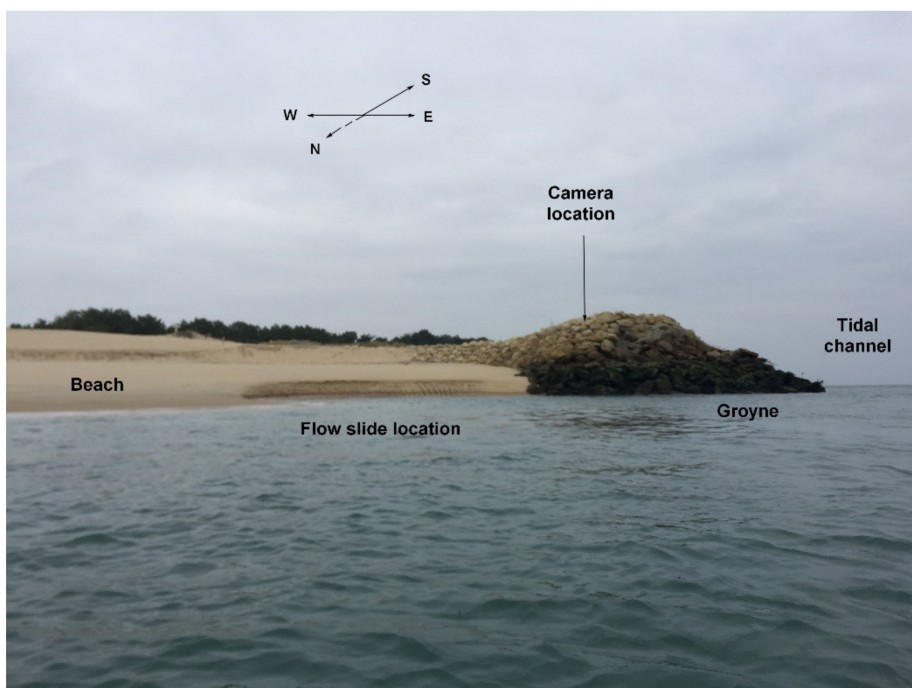

**Figure 4.** Sea view from the sea towards the part of the monitored beach where flow slides occur.

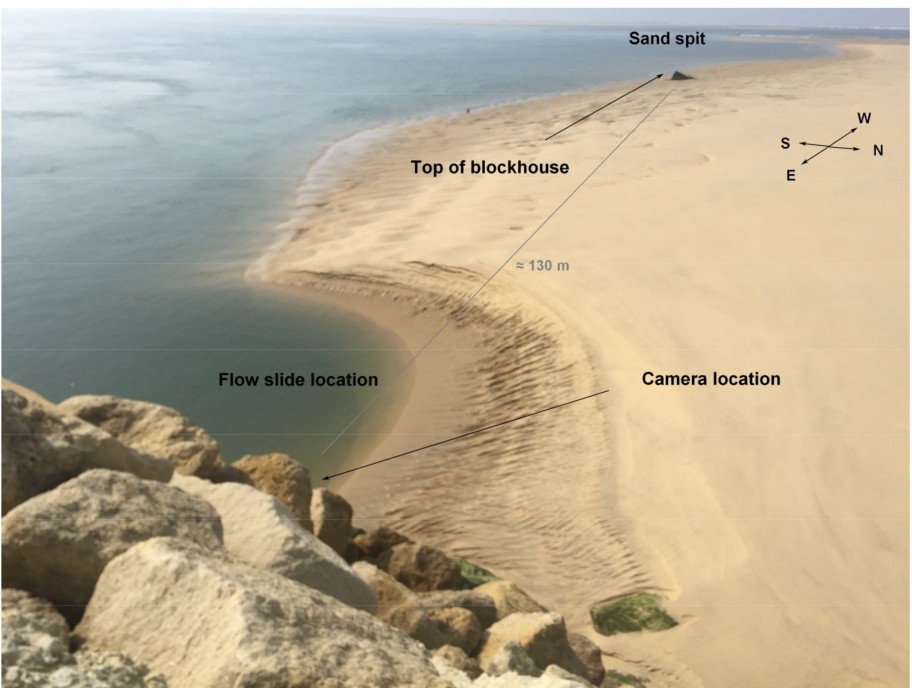

**Figure 5.** View from the top of the groyne towards the whole monitored beach.

### 3. Materials and Methods

*3.1. Experimental Setup*

The experiment was designed in order to meet the following objectives:

- Use simple, low-cost and robust equipment on site;
- Record and timestamp all visible flow slides in an event-oriented chronology;
- Collect a set of still pictures detailed enough to show surface manifestations of flow slide events and allow further work on the influence of environmental factors (images may provide visible factors such as sand features, sea conditions, clouds and rain).

An inexpensive time-lapse camera was chosen as a good compromise between robustness in a coastal environment facing oceanic waves, picture detail and accuracy, and stability of the targeted view. A Brinno 200TLC Pro and its protection case were used and fulfilled the first objective. Resistance to the environment was improved by inserting this into a rock drilled with two perpendicular holes. The first one, closed by a screwed steel plate, allowed the camera to be inserted and secured. The second and smaller hole was open in front of the objective. Image stability was improved too by the mass of the block.

A few steps of this procedure and a view of the camera in place are shown in Figure 6.

Both the block and inserted camera were left in place between 30 July 2015 and 7 April 2017, with a capture interval of 10 min. This interval was a compromise between obtaining details about ongoing events (retrograding slowly) and maintaining long-term monitoring. Camera operation energy was provided by batteries that needed replacement approximately every month. Weather conditions could alter this periodicity, especially because of very high or very low temperatures, impacting data retrieval operations and battery replacement. This operation successively required extraction of the block from the groyne, and an extraction of the camera from the block. After on-site data collection by means of a USB cable, the material replaced.

The monitoring period covered a total duration of 618 days. Unexpected losses of battery charge led to a loss of 42 days; thus, 576 days of observations could be analyzed. The duration of each data gap did not exceed 15 days.

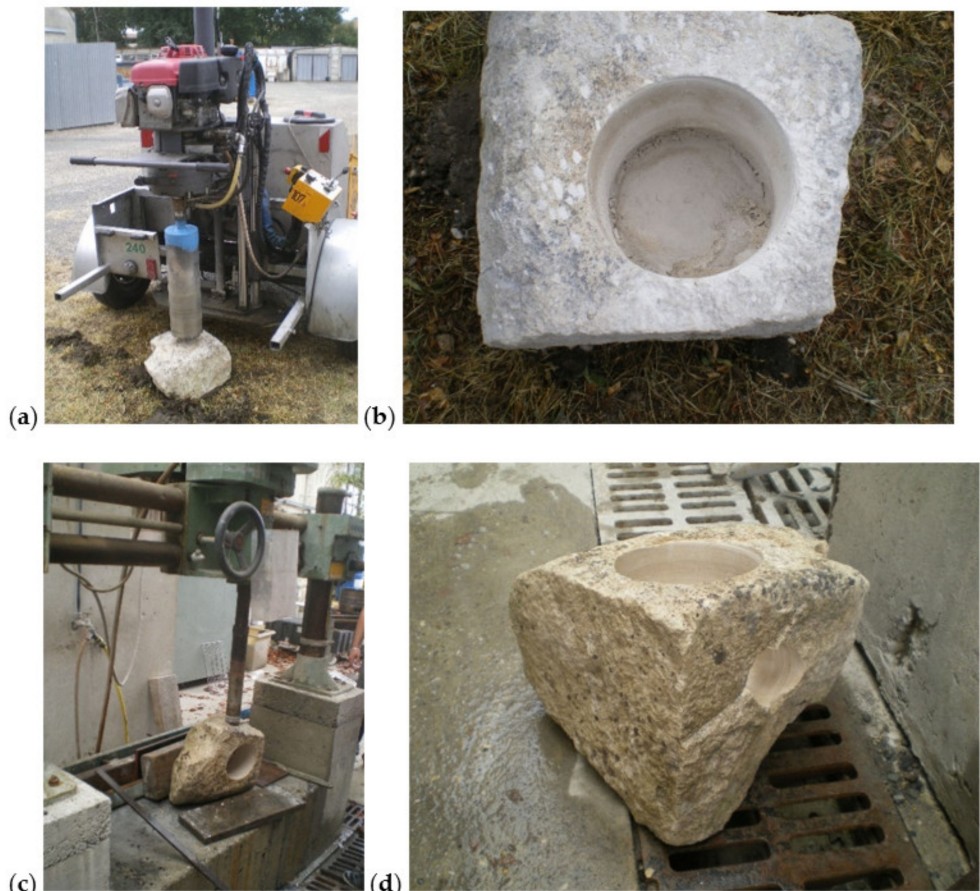

**Figure 6.** *Cont.*

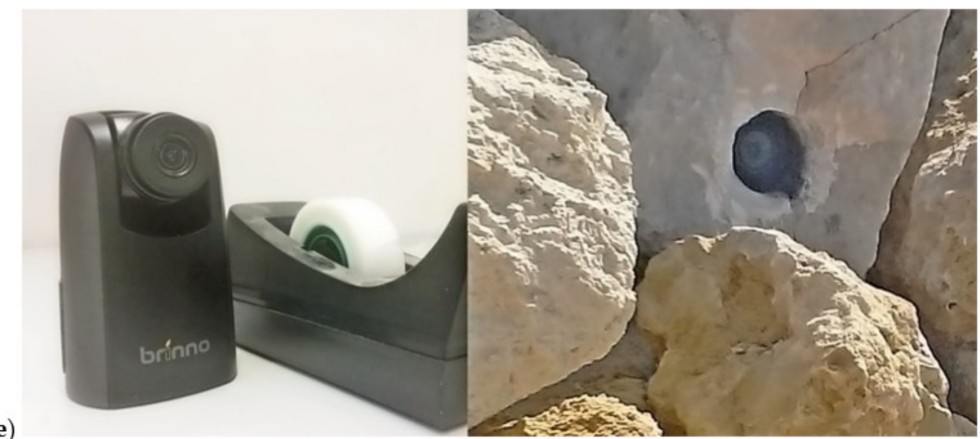

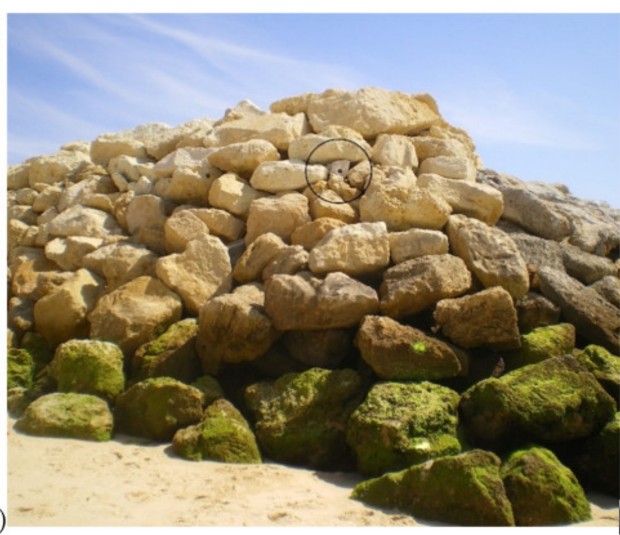

**Figure 6.** Special casing and placement of the camera: (**a–d**) drilling of the block, larger hole and locking plate were placed downside at the site; (**e**) Brinno 200TLC Pro camera, as seen out of package and as can be seen once in the block (large hole and steel plate are downside); (**f**) block inserted in the groyne.

The monitoring data were partitioned into two sets of pictures.

A first large set of pictures was used without geometric processing as a material for the documentation of the main visible characteristics of flow slide progression and beach healing and recovery.

A second set of daily pictures was extracted, and geometrical processing was applied with the purpose of classifying flow slide events and studying longer-term dynamics. These pictures were chosen according to the quality of the image and the water level. The lowest tide level was searched, although it was not always exactly at the low tide time, due to some adverse weather or light conditions.

*3.2. Image Processing*

As specified previously, data recovery entailed block removal from the coastal structure, followed by reinsertion. Due to changes in block position between these two procedures, image processing was necessary to correct them and obtain a fixed field of view (different fields of view in raw pictures are shown in Figure 7). This was performed with Hugin software.

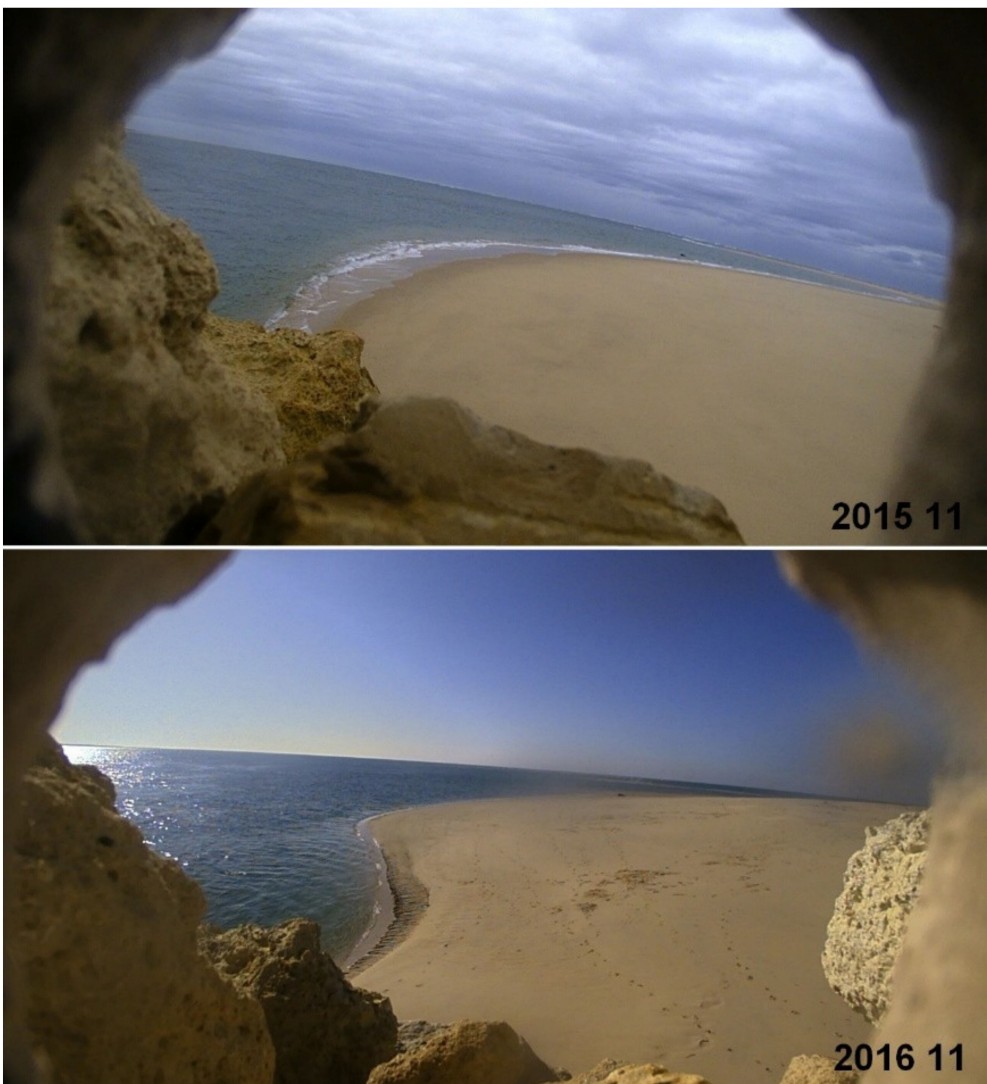

**Figure 7.** Raw views from picture collection periods with two different block positions.

The pictures were processed so that an almost invariant field of view can be seen; thus, differences between block positions were corrected. Hugin software, designed for perspective manipulation, was used to transform views according to a set of two geometric rules: (1) invariable relative coordinate x = 0.6 in the picture of the blockhouse top; and (2) invariant tilting and relative elevation of horizon y = 0.6 in the picture. Image processing by Hugin is carried out through three steps: (1) setting the camera lens parameters; (2) calibration of three Euler angles to satisfy geometric rules; and (3) numerical calculation. The completion of these geometric rules also yielded a quasi-straight horizon. Calibration was performed with two pictures, preceding and following the camera manipulation for data retrieval, respectively. Figure 8 shows the raw and processed pictures. A grid and labels for relative coordinates on the processed picture illustrate the geometric rules.

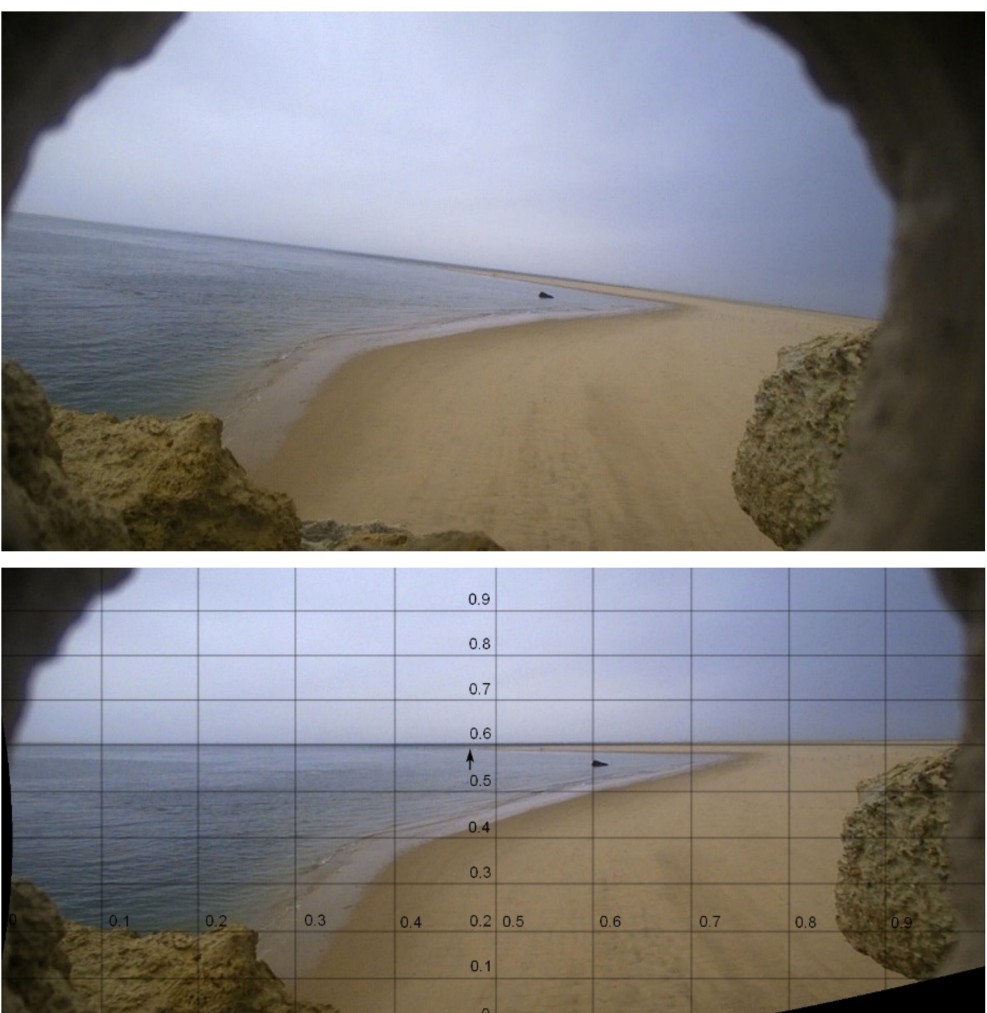

**Figure 8.** Example of a raw picture (**top**) and processed picture with grid and labels with relative coordinates (**bottom**). The black arrow indicates the sand spit end location in the picture (see Section 4.2.2).

Verification of the geometric rules was checked on a selected set of daily pictures with good sight conditions together with low tide conditions (the selection process is detailed in part 3). Observed relative coordinates of the blockhouse corner and the relative elevation and tilting on the picture of the horizon line are shown in Figures 9 and 10. Tilting of the horizon was calculated as the slope of the line bounding two points, on the extreme left and the extreme right of the visible horizon line, respectively (partially hidden by the sand spit on the right). The results were considered to be good considering the overall expected level of geometric accuracy and level of details in close observations.

Vertical dark grey lines marking data retrieval timestamps have been added to both Figures 9 and 10.

Figure 9 reveals the movement of the block and camera during the experiment, due to wave impacts on the groyne. This occurred during the fifth monitoring period and produced a slight change in the blockhouse y coordinate.

Variability of the horizon data can be explained by the picture resolution being too low to ensure a clear view of the background, together with the sensitivity of near-edge pixels to Hugin Euler angles.

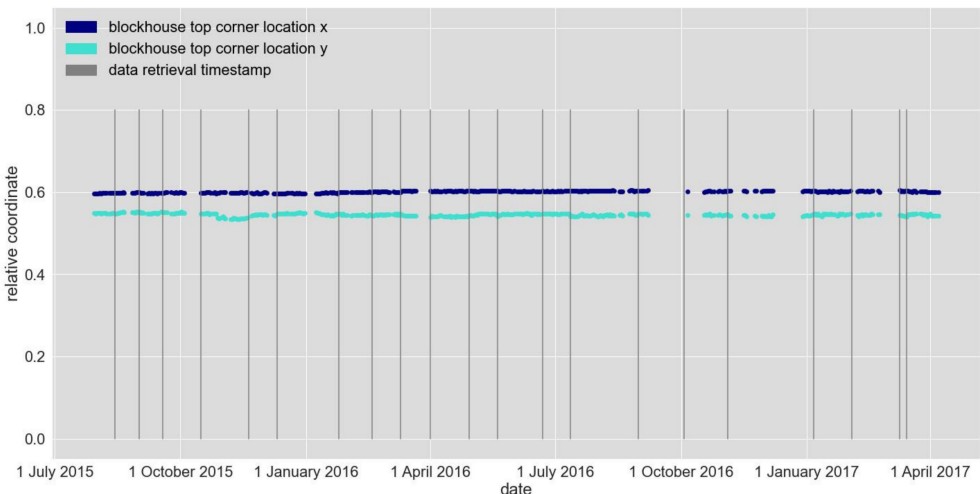

**Figure 9.** Blockhouse relative coordinates in the daily pictures after processing.

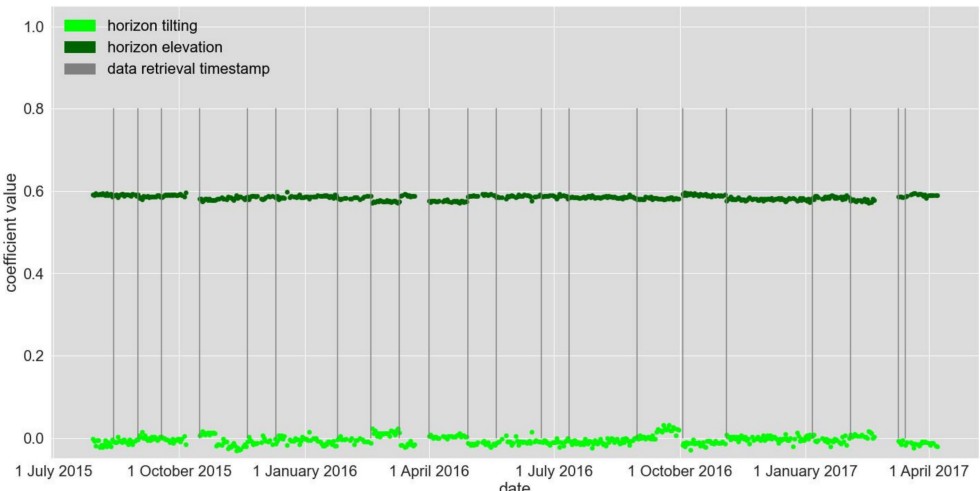

**Figure 10.** Horizon tilting (slope of the horizon line in picture) and relative elevation of the horizon in the daily pictures after processing.

## 4. Results

### 4.1. Main Characteristics of Flow Slide Progress and Beach Healing

A list of all known events was provided by the overall examination of daily pictures, or raw pictures as necessary. Beach changes were qualified as flow slide events when the nearby shoreline was significantly different between two low daytime tides, and revealed a circular scar, either freshly excavated or smoothed by an intermediate nightly tide. A few smaller events needed to be certified after examining sequences of raw pictures with a time step of 10 min. A total of 67 events were found with this method.

Two examples of visible changes at event scale are shown as sequences of successive raw images.

The first example (Figure 11) is that of a small event that takes place at some distance from the groyne (rare) and excavates the beach entirely underwater (barely observed during the day as well). A vortex with a sand load at 8 h 21 revealed underwater slide progress. Increased darkness of the water between 8 h 01 and 9 h 01 revealed the loss of sand below the surface.

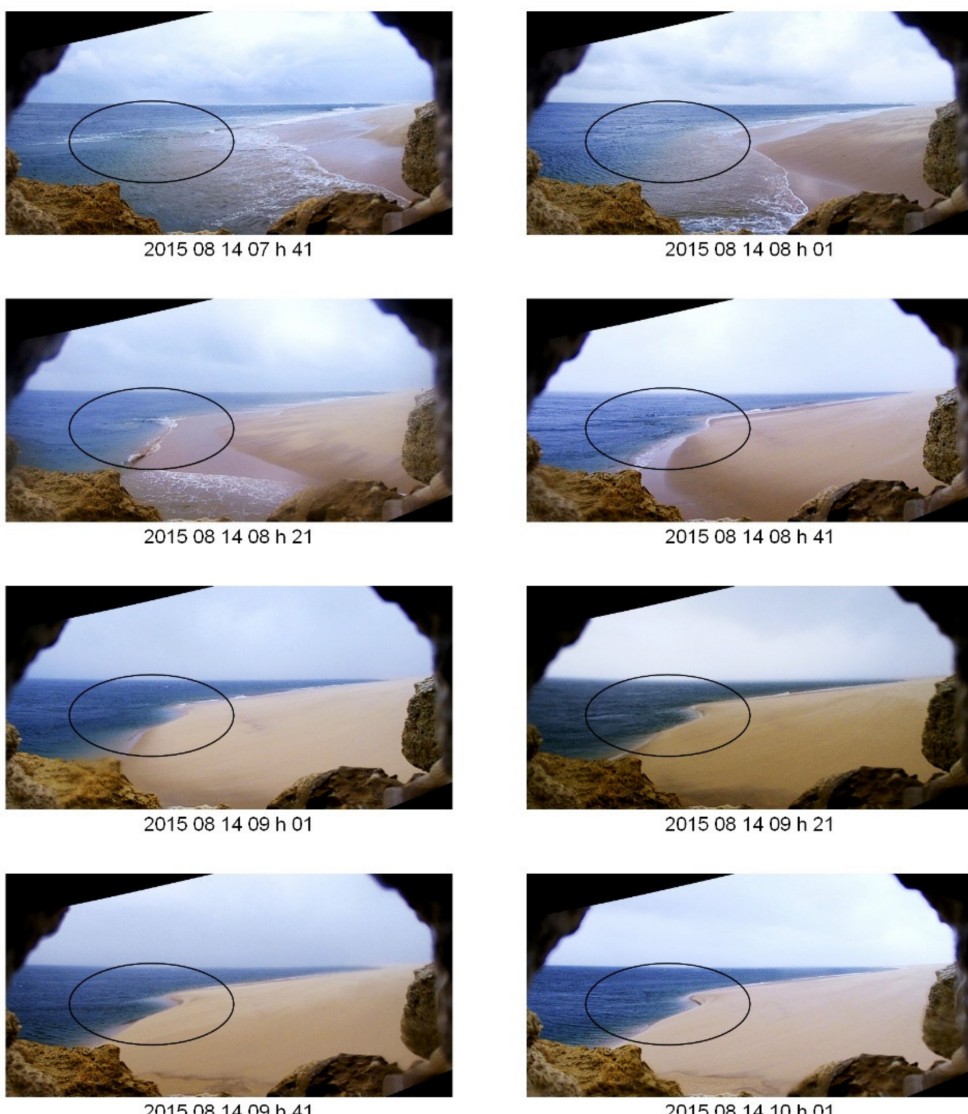

**Figure 11.** Successive raw images showing a mainly submerged event.

The second example (Figure 12) is that of an event that started around noon and progressed into the beach, excavating its emerged part. Foam was present during most of its progression, and was remarkably abundant at 15 h 15 and 17 h 15. The origin of this foam cannot be determined from this set of pictures. Foam is considered by Beinssen [14] to be a characteristic of flow slides in sands which have not been disturbed for some time. The frequent repetition of events here indicates that there may be another cause, possibly with the same foam generation process. Unsaturated sand is suspected to produce foam; however, the smoothed excavation rim after 15 h 55 and an invisible part of the progressing event suggest alternative possible explanations. Large sand plumes emitted by the sliding phenomenon can be seen at 16 h 55. Pictures from 14 h 15 to 15 h 15 give a good idea of the rapidity of the progress.

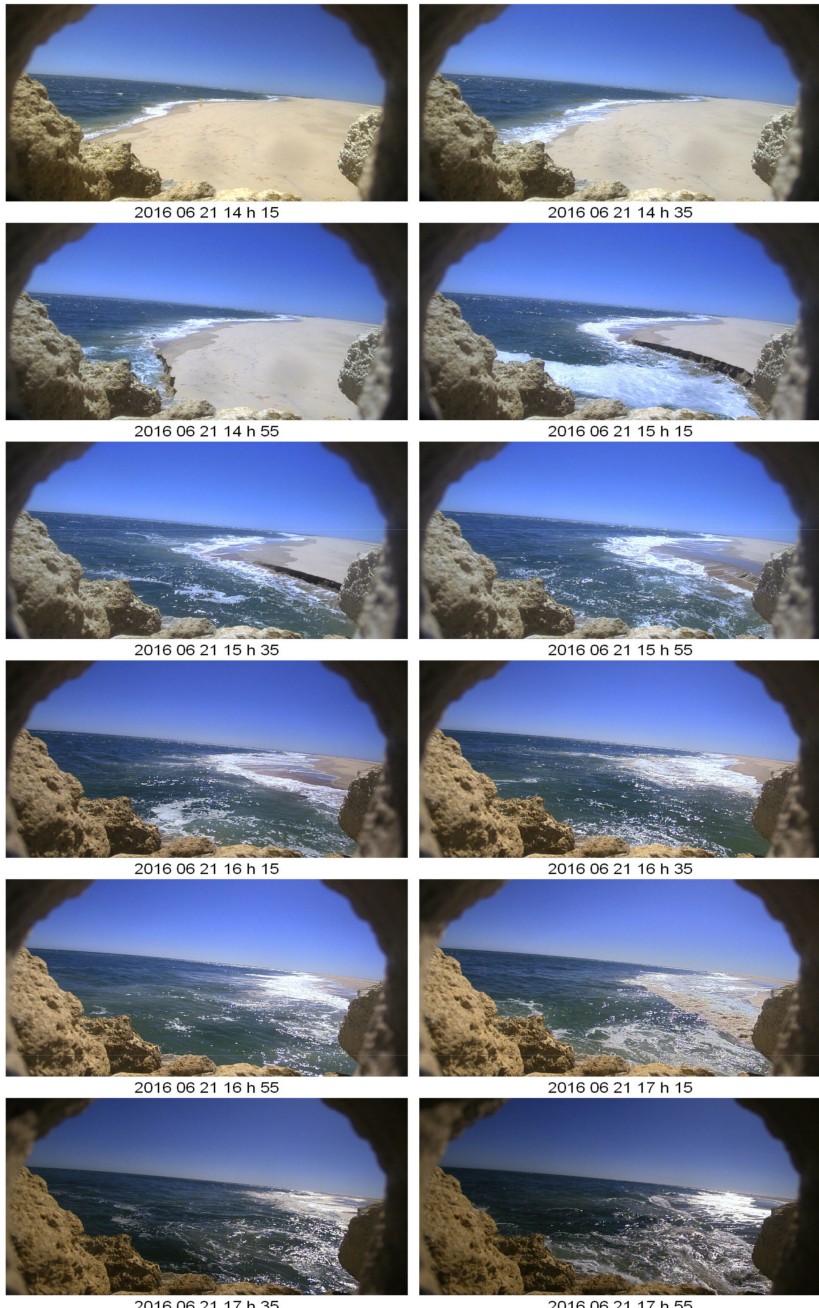

**Figure 12.** Successive raw images of an event excavating the emerged part of the beach.

Daily pictures present interesting information about the beach healing process. This process depends on drift intensity; flow slides and coastal dynamics are closely related. Figure 13 shows an active succession of slides (Dates of these slides: December, 14, December, 15, December, 17, December, 20, 2016) and beach healing, which demonstrates the ability of the longshore drift to fill the void left by large flow slide events. During such periods of intense activity, the top corner of the blockhouse is covered by sand, as will be explained in Section 4.2.2.

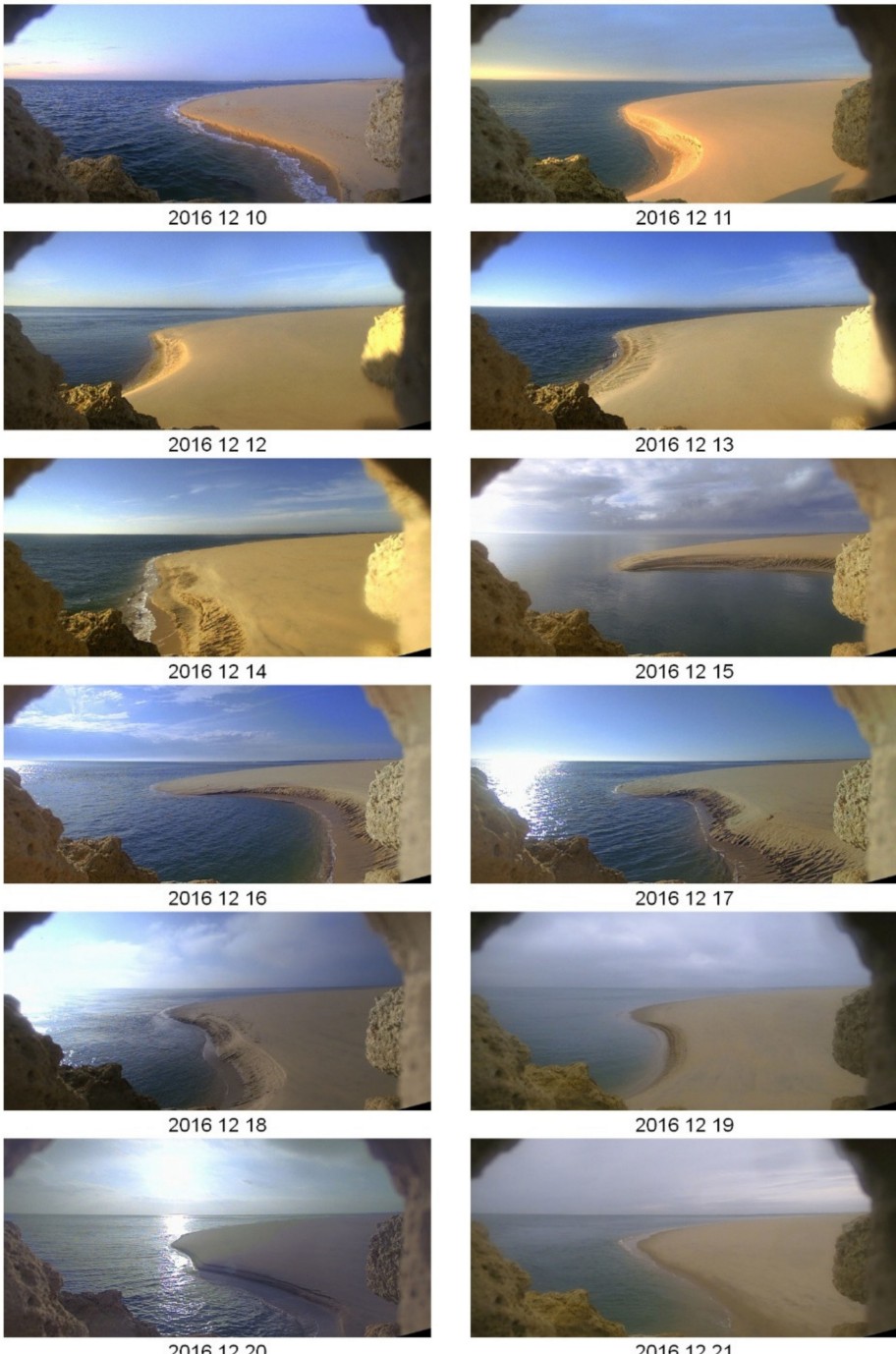

**Figure 13.** Daily processed images showing a succession of flow slide events and subsequent healing periods.

This sequence of images also shows that flow slides may occur before total healing from the preceding event.

The next section will present that the flow slide frequency, influence of drift and healing capacity are not constant between seasons and sedimentary conditions.

*4.2. Long-Term Flow Slide Dynamics and Environmental Influences*

The collected pictures revealed large variability in slide triggering and size of the beach excavation; thus, a more time-related approach was sought in order to characterize the flow slide dynamics. Two attempts for event classification by size were performed as a

preliminary assessment, prior to the further discussed assumption that the event size may be a relevant indicator of longshore sediment transport.

The next section introduces the simple and improved classification methods based on event intensity. The subsequent, final section introduces an application for the study of environmental influences through the relationship between changes in the western sand spit and the flow slide regime.

### 4.2.1. Classification of Breaching Flow Slide Events

The whole set of images gave the possibility to identify and characterize flow slide events. However, it also revealed a great variety of excavation shapes, sizes and locations. Therefore, it was first attempted to categorize these events between distinct classes, according to magnitude. Scant quantitative data could be drawn from pictures, and the visible shape appeared as the only criterion. Time variability was unknown as well. A group of five classes was chosen at first, and an operator distributed events between them, basing their choice on aspect similarity. One event was observed as significantly larger than the others, and was separated in a sixth class.

Although a visual examination of raw pictures allows a simple evaluation of crater sizes, and consequently, of event intensity, an alternative method was established for a more systematic and quantitative evaluation, in order to address its severity and the danger caused by the excavation nearshore as well as within the beach. The adopted quantitative indicator was the alignment of the shore near the groyne, broken by the excavation in cases of flow slide. This broken alignment can be measured as an angle formed by three points on the shoreline.

The sets of three points were digitized on images with a script developed in Python. Figure 14 shows a group of views captured from the digitizing process. Three sets of points (in blue, purple and red) visualized shoreline changes between former daily views.

A red star on the left indicates that the picture was taken the day following a flow slide event. In this case, one triplet was digitized on the distant corner side of excavation. Following triplets were digitized close to groyne again, even during the beach healing process. If a flow slide excavated an unhealed former event, one triplet was digitized on the new excavation corner.

Angles associated with triplets are meant to evaluate the shoreline deflection, substantially affected by excavations. The following schematic (Figure 15) illustrates the measurement principle. According to these principles, most angles should be close to 0, whereas angles related to flow slide events should have greater values, reaching more than 90 degrees.

Angles and events are presented on a timeline graph split into two periods (Figure 16). Missing dates are marked with a dotted line. Events are marked with red dots added to the curve. They mainly appear on peaks, but may be located on other parts of the curve when two events succeed one another before complete healing.

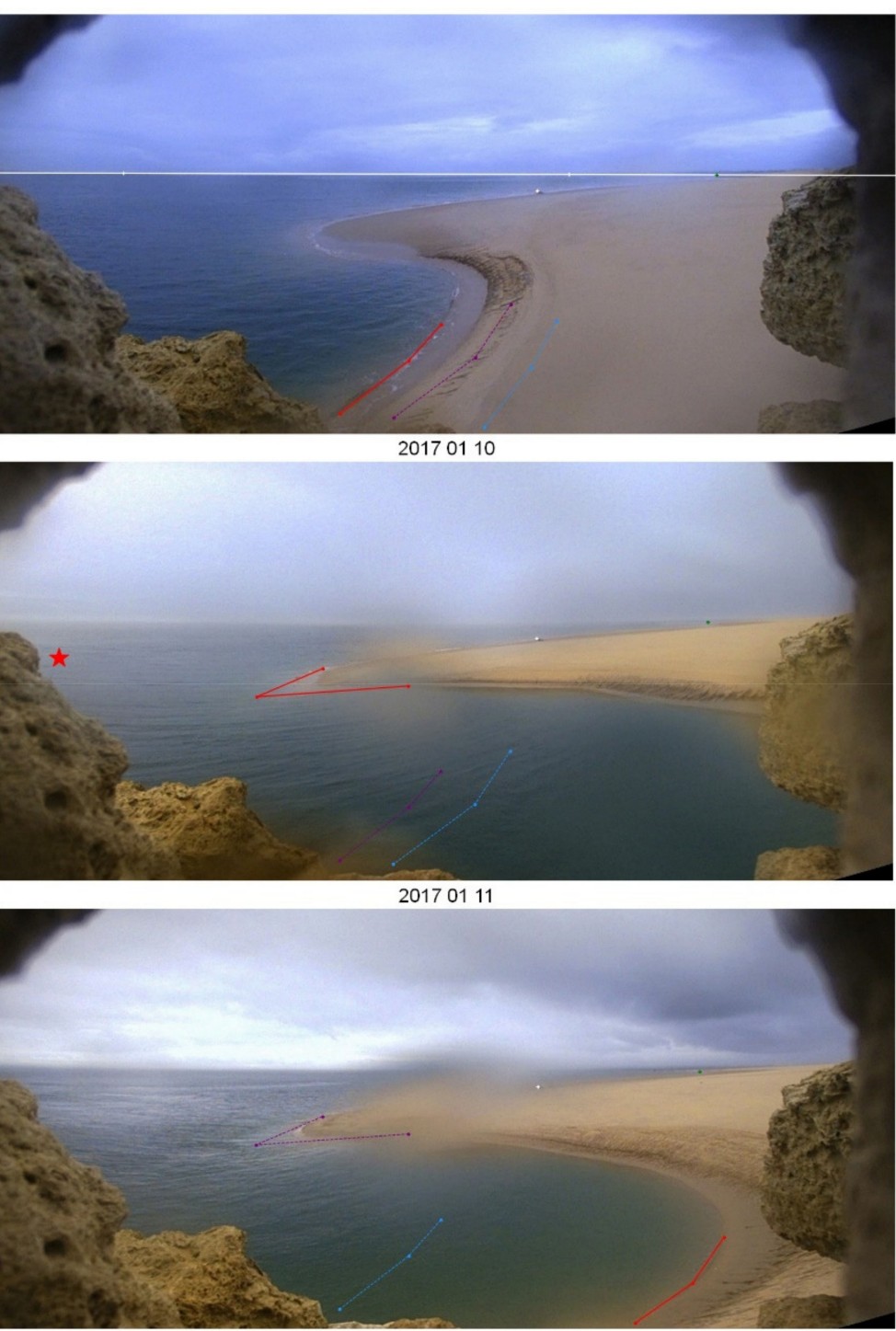

**Figure 14.** Sequence of views captured on the screen during the digitizing process. Blue and purple lines are previous digitized shorelines, and the red line indicates the shoreline associated with the current view. Red stars mark events.

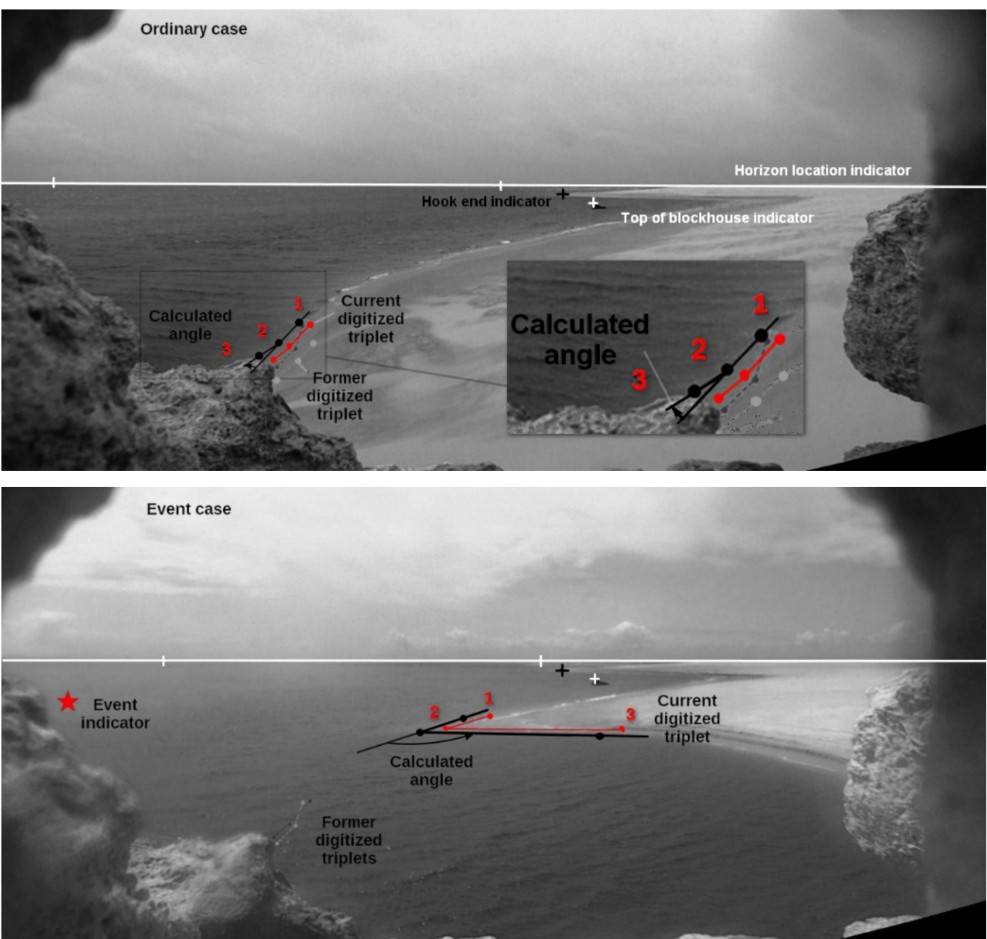

**Figure 15.** Schematic of the event intensity measurement based on the calculation of the nearby shoreline angle: ordinary case (**top**) and event case (**bottom**). Red dots and lines: current triplet used for angle calculation; black dots and lines: shifted copy of current triplet with graphical illustration of calculated angle; light grey and dark grey dots and lines: former locations of triplets.

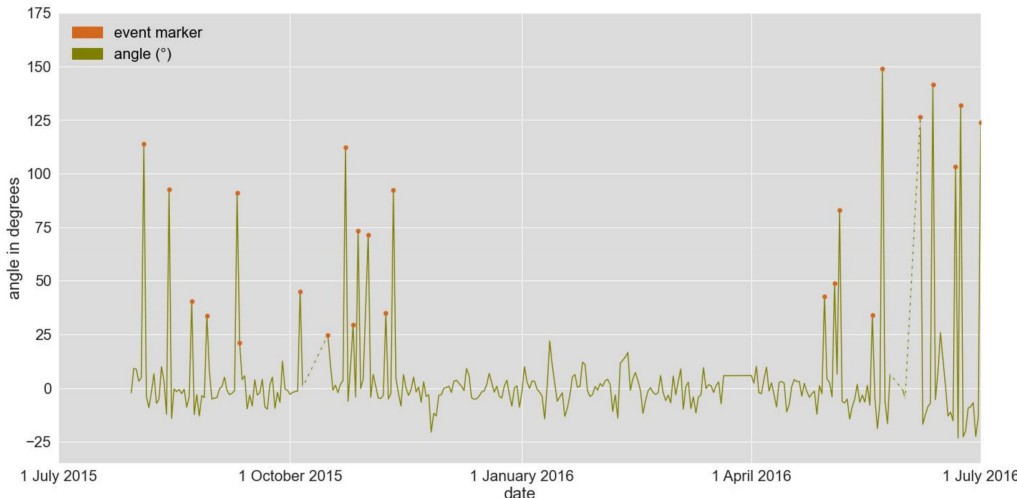

**Figure 16.** *Cont.*

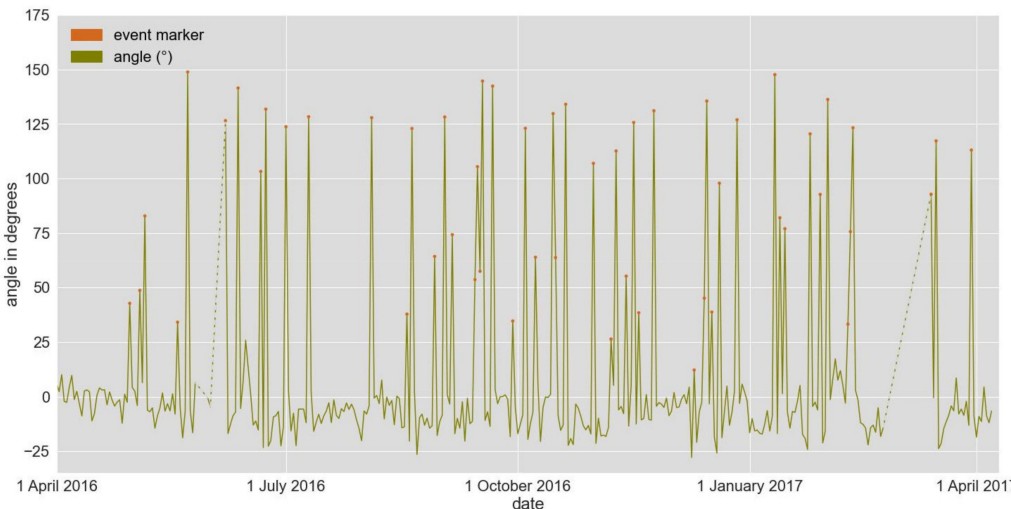

**Figure 16.** Timeline of shoreline angle and event markers.

This quantitative method was compared with the simple visual classification of events into six classes. Figure 17 presents the angle versus the visual class. The only event in the sixth class was left separate because of its size being slightly greater in appearance than that of the largest class. This chart shows adequate overlapping between ranges of angles and visual classes for the first four classes. However, the operator's human eye could distinguish differences between events with angles greater than 120 degrees, which could not be distinguished by the angular method. This may be related to the loss of picture accuracy with distance and exact shoreline shape identification. In the Discussion, it is presented that this difference may not significantly affect season-scale interpretation.

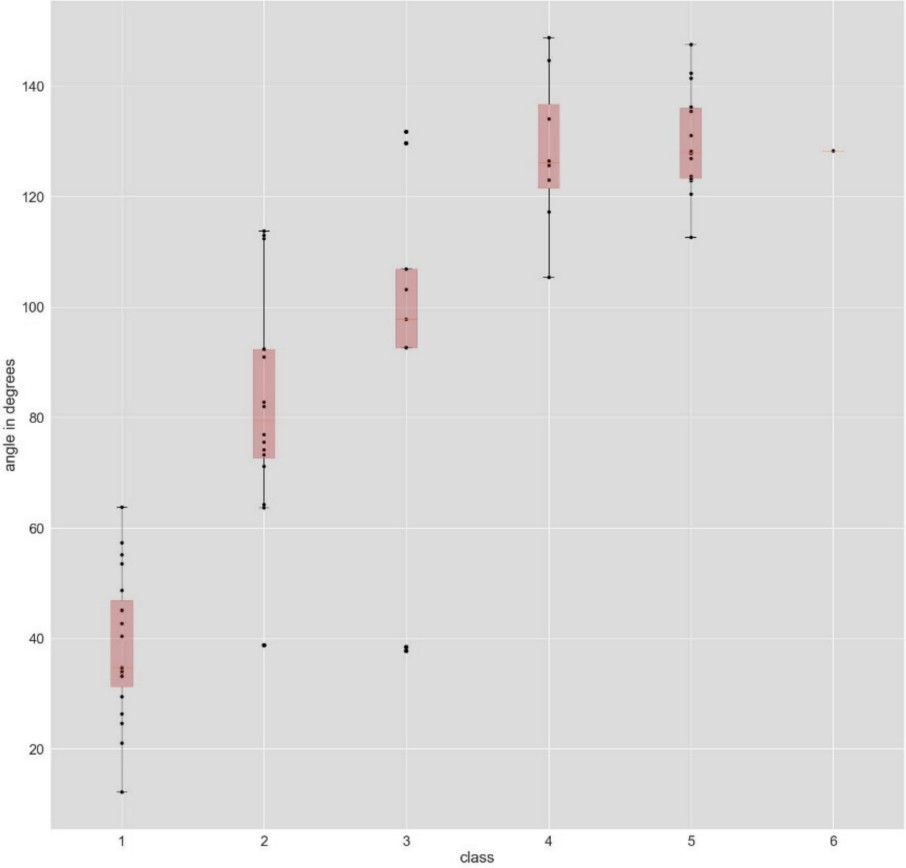

**Figure 17.** Comparison graph of event angle value versus visual class.

Differences between the visual class and angle value are illustrated by the four following representations of an image with their corresponding point in a comparison graph (marked by a red dot) (Figure 18). This shows the discordance between both methods due to double event occurrence, or a lack of accuracy with distance, in particular.

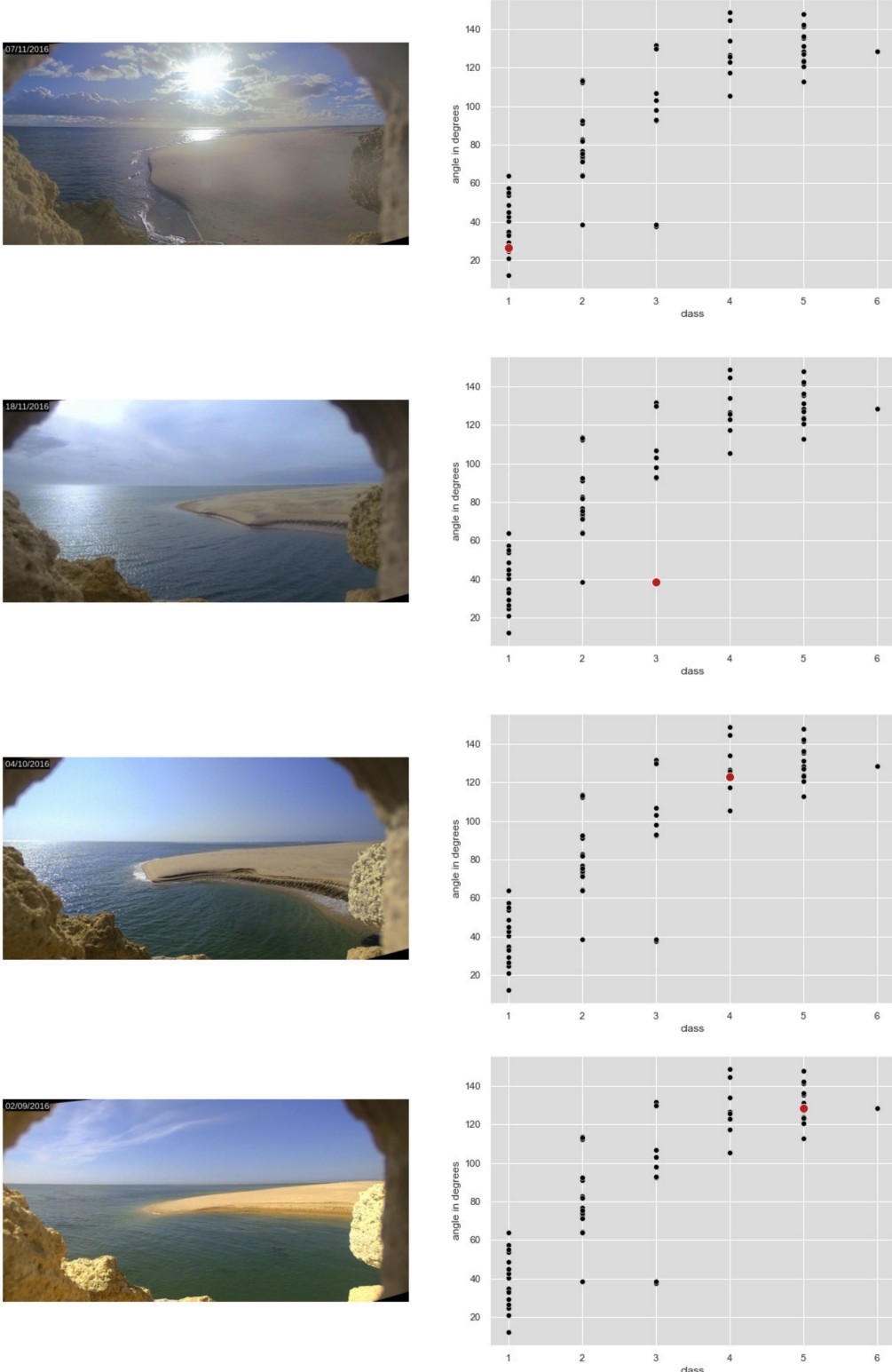

**Figure 18.** Four examples of event views together with comparison graphs. The corresponding point is red.

4.2.2. Application to a Study of Environmental Influences

An examination of daily images revealed that the end of the visible sand spit, moving southward and then regressing northward, might be related to changes in the flow slide regime. Figure 16 shows a dramatic cease in flow slide activity between November 2015 and May 2016. During this period, the sand spit was massive, whereas the beach was in a very flat and rippled shape, as shown in Figure 19. The southward extension of spit was particularly visible on 19 February 2016.

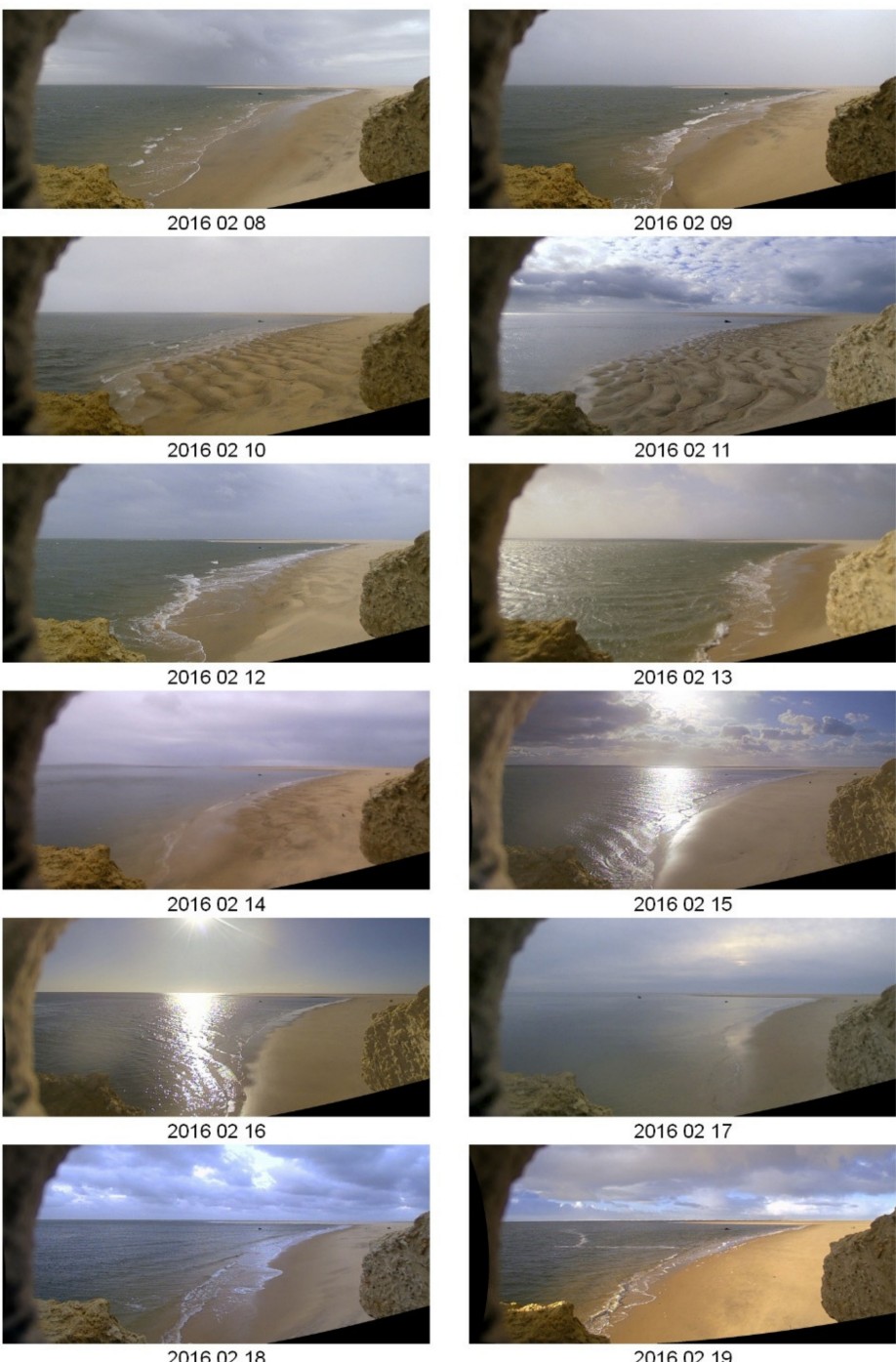

**Figure 19.** Sequence of daily pictures during low flow slide activity.

Locations of the end of the western sand spit in daily pictures (see Figures 5 and 8) were recorded in order to study its influence. For example, in Figure 8, the relative coordinates

of the spit end are (0.47, 0.59). All daily relative coordinates are plotted on the graph in Figure 20. Despite the loss of accuracy due to distance- and calibration-related uncertainties, changes in the x value were considered as good indicators of a southward extension of the spit. This was simultaneously observed through topographic surveys by Nahon [17].

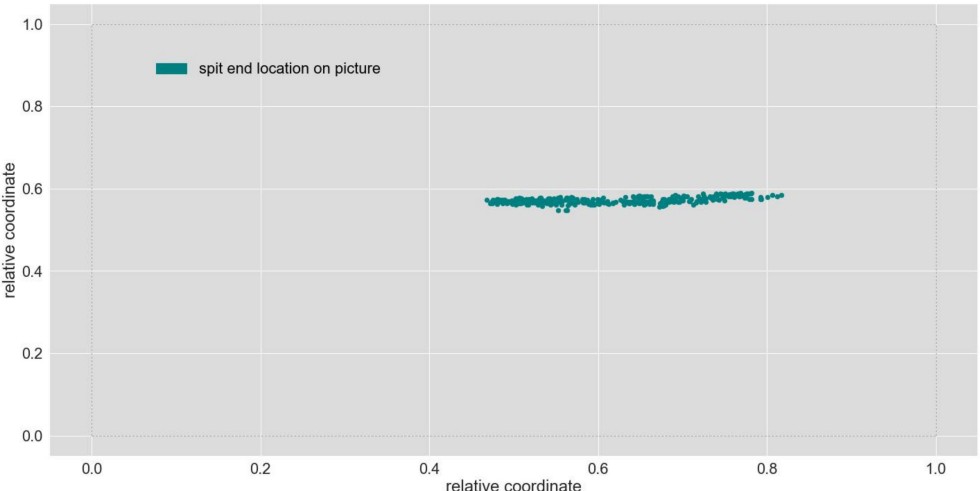

**Figure 20.** Relative locations of western sand spit end in daily pictures after calibration.

In the subsequent Discussion section, we develop an analysis, using the monitored data, of the relationship between the interruption of flow slide mechanisms, as seen in Figure 16, and movements of the sand spit, as seen in Figure 20.

## 5. Discussion

The image collection featured in this paper provides interesting knowledge about the flow slide phenomenon on a site where it occurs frequently in a controlled area. The method we introduced is a relevant survey technique considering the main characteristics of flow slide and beach healing, as well as time-related aspects of event dynamics. It also opens the way to programs including visual monitoring together with physical data acquisition.

### 5.1. Analysis of the Relationship between the Flow Slide Dynamics and Movements of the Sand Spit

Highlighting the relationship between sand spit changes in shape and flow slide activity was performed following a method developed for artificial drainage analysis: drainage discharge, characterized by a peak regime, is summed into a cumulative dataset and plotted as a function of rainfall or accumulated rainfall. This approach is of much help in studies considering environmental impacts on groundwater drainage [18,19]. Flow slide event angles were summed on the same principle and presented as a function of time, together with relative spit length along the x coordinate proportionally fitted between 0 and 1 (i.e., $(x - x_{min})/(x_{max} - x_{min})$). The sum of the visual classes was also plotted on a graph. This result is shown in Figure 21: blue dots indicate the evolution of spit length. Cumulative event intensity is drawn as two normalized curves of values, one for each classification method. Values range between 0 and 1. By doing so, both interpretations of curves appear to be convergent, despite differences between the classifications of larger events. The halt in flow slide dynamics can be precisely dated with this approach.

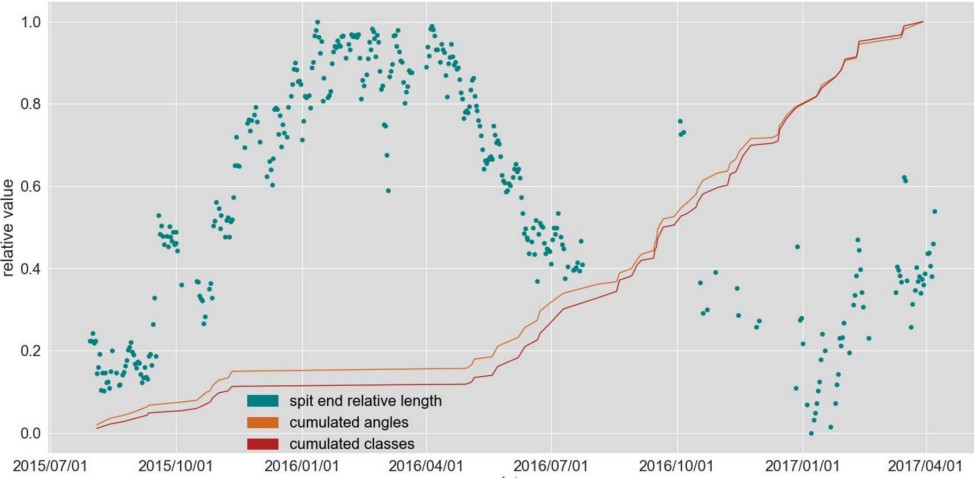

**Figure 21.** Timelines of cumulative sum of flow slide event angles and classes together with relative spit end lengths along the x coordinate.

The slope of the curve indicates a relatively regular time distribution of flow slide intensity. A relationship between this slope and average sand flux through longshore drift may be assumed, if the volume of moved sand during each event can be associated unambiguously with each angle or class. This may be a goal for a future study including precise topographical surveys (see Section 5.2).

The graph in Figure 21 shows the concomitance between the ceased flow slide activity and end of the sand spit extending southwards. It can be seen qualitatively that after this maximum extension, sand accumulated at the spit moved towards the groyne, and refueled the flow slide regime. This regime appeared to be fairly constant during the experimental period.

This application shows true potential for studies of relationships between sedimentary processes and the occurrence of flow slides, as recorded in Figure 16 and classified in Figure 17.

The results obtained are still insufficient to assess sand transport or the influence of non-visible factors. Hence, there is a need to compare and supplement these results with other data acquisition methods.

### 5.2. Monitoring of Breaching Flow Slides Combining Visual and Physical Data

The method described above should be compared with other existing techniques based on the capture, processing, and analysis of video or time-lapse images. Similar qualitative or quantitative methods dedicated to coastal changes are reviewed in the introduction of [20], as well as in [21]. Some techniques have been specifically developed to investigate or detect coastal processes similar to those presented in this paper: instability and failure triggering [22], cliff erosion [23], and sand feature changes and longshore sediment transport [24–27].

The visual monitoring of an unstable cliff described in [22] indicates that the combination of frequently acquired digital images with local displacement measurements can provide useful information regarding the evolution of a rock cliff. A similar association of image collection described in this paper with the medium- or high-frequency acquisition of environmental parameters could yield interesting information as well.

Weather and sea conditions may be influent factors. An overview of sea level and roughness can be drawn from picture examinations. However, event dynamics do not show many differences between summer and winter conditions. Flow slide triggering can probably be better explained in terms of head gradient from the beach groundwater to channel waterbody, leading to submarine ground discharge [8]. Measurements of other varying parameters, such as wind speed, wave height, atmospheric pressure, groundwater

level at the beach, or rainfall, would therefore be helpful too. All these parameters need continuous recording, and their collection represents a project of some importance. Alternatively, some indicators easily brought by visual data (sea roughness or wave runup classes, for example) could be studied similarly to the methods presented in Section 5.1.

General knowledge on the course of flow slides distinguishes between triggering at the toe and excavation rate and retrogradation related to beach topography [8]. Therefore, additional topographic surveys could be conducted to answer the questions about triggering and cross-shore progress. Other methods based on image processing include photogrammetry to collect topographic data [25,27]. However, at this site, photogrammetry would have been impractical because of beach surface uniformity and the difficult use of permanent or frequently laid down ground markers.

Indications on geo-mechanical or hydraulic factors are necessary too [15,28]. Recent investigations carried out at Amity Point [4] raise an open question about the role played by sand compaction in the vicinity of sliding places. Adding a penetrometer field campaign [29] to topography surveys will be helpful on Cap Ferret, considering the great variety of beach shapes observed before slide events. Analysis of the combination of beach slope, together with sand state of compaction and possible erosive currents at the toe, may be important too, considering the high velocities observed in the tidal channel along coastal defenses. Measurements of water velocity in the vicinity of a groyne can thus be included within an overall appropriate monitoring program.

Collected pictures show that calm sea conditions and clear water may take place during a flow slide event. Dynamics of flow slides are now better known on this site; therefore, there is an interesting research opportunity to assess the physical aspects of underwater flow slide processes. In particular, suitable experimental conditions can be met for submarine observations and comparisons with large-scale laboratory experiments [10] or controlled on-site experiments [15].

## 6. Conclusions

The method developed on the Cap Ferret site gave interesting results about breaching flow slide dynamics, through an unusual chronological point of view. It represents a good approach for the study of seasonal influences, because it revealed connections with longer-term sedimentary processes. Although limited to the daytime, collected data and information will add valuable knowledge to studies and research still engaged at other sites [8].

These sets of pictures and flow slide event chronology open further research perspectives on how weather and sea conditions influence sedimentary processes at smaller scales, and may influence studies about longshore sand movements in interactions with flow slide events, such as estimations of post-event displaced sand volumes, interactions between unstable area and morphological evolutions of sand spits and southern beaches, considering longshore drift along western beaches.

**Author Contributions:** Conceptualization and methodology, Y.N. and C.G.; field instrumentation and data collection, P.F.; picture processing, F.C. All authors have read and agreed to the published version of the manuscript.

**Funding:** This research received no external funding.

**Institutional Review Board Statement:** Not applicable.

**Informed Consent Statement:** Not applicable.

**Data Availability Statement:** Not applicable.

**Acknowledgments:** The authors would like to thank Benoît Bartherotte, owner of the coastal defense and groyne, for allowing access to the site during the experimental setup and data retrieval sessions.

**Conflicts of Interest:** The authors declare no conflict of interest.

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
