# Peer review of "Time-Lapse Camera Monitoring and Study of Recurrent Breaching Flow Slides in Cap Ferret, France"

_2673-964X, doi:10.3390/coasts2020005_

Round 1

Reviewer 1 Report

Overall, the manuscript is clear, concise, and well-written. The methods that are used to analyze the data are appropriate. The presentation of the introduction, results and discussion are satisfactory. The study and objective of the research are very relevant and important in current situations. However, I have a couple of minor suggestions that I believe should be addressed before the paper is published.

I miss some relevant literature from the work, e.g., Ahn et al., 2017. J. Coastal Res., 79, 204–208. and Alhaddad et al., 2020. J. Mar. Sci. Eng. 8, 67.

Figure 1. Please use also the English name of the bay in the figure and in the caption (Arcachon Bay).

Figure 3. In Cap Ferret, there is a groin field along the shore to the north. Better indicate as "Groyne field" in the figure. Do not use "sandy hook" in the figure. Use "sand spit" such as in the caption.

I would recommend adding a couple of sentences about the limitations of the technique in the conclusion section.

Reviewer 2 Report

Dear Authors,

I really appreciate your work, well written and quite interesting. However, after a thorough review, in my opinion, the paper has some shortcomings regarding the analysis of the results.
I have provided my remarks in the attached file.

Finally, I suggest that Discussion and Conclusion should be separated. In the Discussion chapter, you notice essentially the limits of your method (e.g., lines 333-338, and 358-371), scarcely highlighting its potentiality and poorly discussing the obtained results (e.g., the occurrence of the events in Fig.16 and the weather conditions, the meaning of the event angle values and the visual classes of Fig.17). Moreover, Fig.21 should be better explained and could be potentially adopted for completing the Discussion.

Lines 378-385 should be integrated in the Conclusion section.

Round 2

Reviewer 2 Report

Dear Authors,

thank you for considering my suggestions; I really appreciate the increasing quality of the results' discussion and the overall improvement of the paper.